# The Potential for Bio-Sustainable Organobromine-Containing Flame Retardant Formulations for Textile Applications—A Review

**DOI:** 10.3390/polym12092160

**Published:** 2020-09-22

**Authors:** A Richard Horrocks

**Affiliations:** Institute for Materials Research and Innovation, University of Bolton, Deane Road, Bolton, Greater Manchester BL3 6HQ, UK; a.r.horrocks@bolton.ac.uk

**Keywords:** bromine, organobromine, flame retardant, textiles, environment, sustainability, antimony III oxide, zinc stannate, zinc tungstate, smoke suppression

## Abstract

This review considers the challenge of developing sustainable organobromine flame retardants (BrFRs) and alternative synergists to the predominantly used antimony III oxide. Current BrFR efficiencies are reviewed for textile coatings and back-coatings with a focus on furnishing and similar fabrics covering underlying flammable fillings, such as flexible polyurethane foam. The difficulty of replacing them with non-halogen-containing systems is also reviewed with major disadvantages including their extreme specificity with regard to a given textile type and poor durability.The possibility of replacing currently used BrFRs for textiles structures that mimic naturally occurring organobromine-containing species is discussed, noting that of the nearly 2000 such species identified in both marine and terrestrial environments, a significant number are functionalised polybrominated diphenyl ethers, which form part of a series of little understood biosynthetic biodegradation cycles.The continued use of antimony III oxide as synergist and possible replacement by alternatives, such as the commercially available zinc stannates and the recently identified zinc tungstate, are discussed. Both are effective as synergists and smoke suppressants, but unlike Sb_2_0_3_, they have efficiencies dependent on BrFR chemistry and polymer matrix or textile structure. Furthermore, their effectiveness in textile coatings has yet to be more fully assessed.In conclusion, it is proposed that the future of sustainable BrFRs should be based on naturally occurring polybrominated structures developed in conjunction with non-toxic, smoke-suppressing synergists such as the zinc stannates or zinc tungstate, which have been carefully tailored for given polymeric and textile substrates.

## 1. Introduction

Organobromine-based flame retardants (BrFRs) for bulk polymers and textiles are generally considered to be the most abundantly used after aluminium trihydrate. While the world market for and use of flame retardants continues to grow, the environmental pressures on BrFRs, however, especially during the last 10 years, have been such as to see their gradual replacement by phosphorus- and/or nitrogen-containing species in the main, although recently, the potential environmental toxicity of selected organophosphorus-based retardants has come under scrutiny [1]. It is likely, therefore, that in the foreseeable future, their share will decline.

Since the late 1980s, when the issue of dioxin formation during the incineration of polymers containing bromine-containing flame retardants first arose [2,3], an issue which is still current [4], all halogenated retardants have been under scrutiny based on a number of significant identified potential toxicological and environmental impacts.

Without wishing to enter into extreme detail, following the initial concerns in Germany in 1986, the EU published in 1991 a draft amendment to EC Directive 76/769/EEC, which would essentially ban use of all polybrominated diphenyl oxides or, as they are more recently referred to, polybrominated diphenyl ethers (BDEs) within five years. In 1994, this Directive was withdrawn as subsequent studies cast doubt on the earlier concerns. Simultaneously, other organisations (e.g., US Environmental Protection Agency, OECD, EU) initiated risk analyses of these compounds. In addition, the World Health Organisation initiated an evaluation of the risk to health of BDEs, which in 1994 indicated that they did not pose a significant hazard. However, during the next 25 years, the focus of concerns changed to their potential toxicological and ecotoxicological effects, particularly of the polybrominated diphenyls and diphenyl ethers, leading to effective bans in the EU of penta- and octa-bromo diphenylene ethers of the production and use of the former in 2002 and voluntary phasing out of both in the USA in 2005; they were listed as persistent organic pollutants or “POPs” by the Stockholm convention in 2009 [5,6,7].

During the past 15 years or so, principal environmental concerns have included bioaccumulation, which has been observed across the world’s continents [8,9,10,11] including the Antarctic [12] and Arctic regions [13], and the degradation into ecotoxicological congeners in the case of polybrominated diphenyl ethers by exposure to UV [14,15,16] and microbiological agents [8,17,18], for example.

It must be remembered, however, that bromine-containing flame retardants are only effective when used together with a synergist of which antimony III oxide (ATO) is the most important [19]. In fact about 50% of the annual production of ATO (~100,000 tonnes in 2017 [20]) is used as a synergist for use with halogen-containing polymers like poly(vinylidene chloride), poly(vinyl chloride) and their copolymers [21] and as well as halogen-containing flame retardants. However, ATO has known hazards of lung toxicity, lung cancer, delayed ossification and maternal toxicity and its GHS (Globally Harmonized System of Classification and Labelling of Chemicals) classifications are H351 potential carcinogen class 2 and H373 STOT RE lung (voluntary listing, STOT = specific organ toxicity) [22,23]. While its use is easily controllable in the workplace during formulation and application and once in a coating or bulk polymer it is fixed, it generates end-of-life challenges. Its presence is often about 50% by weight with respect to the flame retardant species present and so comprises a significant proportion of any product comprising a BrFR formulation.

## 2. Organobromine Flame Retardants—Environmental Challenges

The popularity of bromine-containing flame retardants, especially within the textile sector, the focus of this review, is a consequence of a number of advantageous factors:The high atomic weight of bromine, which ensures that BrFR molecules comprise typically >60% bromine, thereby reducing the total amount of flame retardant required.Acceptable levels of flame retardancy can usually be achieved with bromine levels of 5–10 wt%, which enables total BrFR contents to be in the range 9–15 wt%, much lower than for many phosphorus-containing species in which phosphorus contents are often <30 wt%.The carbon-bromine bond is strong enough to resist normal processing temperatures typically up to 250 °C or so, especially if it is an aromatic C–Br bond, while it breaks down into active bromine^.^ radicals above 300 °C.

While the chemical forms and types of BrFRs in use are many and varied [24], they may be simply categorised physically in terms of whether they are monomolecular liquids (usually only when phosphorus is also present, e.g., tris(1-chloro-2-propyl) phosphate (TCPP)) with obvious mobility issues, monomolecular solids having increased insolubilities and intractabilities and polymeric forms, which may comprise polymeric additives, comonomers or bromine-containing polymers. This review will focus in the main on those with applications to the textile area, which most usually comprise coatings or back-coatings [25], although in the case of polypropylene fibre-containing fabrics, bromine-containing melt-additives such as tris(tribomoneopentyl) phosphate are often used [26]. In the UK for the last 30 years or so, such treatments have assumed the major share of both domestic and contract furnishing fabrics. However, it must be noted that unlike the commonly used BrFRs, because tris(tribomoneopentyl) phosphate contains both bromine and phosphorus, it is not accompanied by antimony III oxide as synergist, although small amounts of hindered amine stabilisers are reported to enhance its effectiveness in polypropylene fibres [26].

The majority of fabrics used as outer fabrics and lining or barrier fabrics in UK furnishings and furniture that conform to the flammability requirements of the UK regulations [27], do so by the application of a back-coating comprising a BrFR in combination with the synergist antimony III oxide (ATO) and a binding resin. Principal BrFRs used until recently included the brominated diphenyl ether, decabromodiphenyl ether or DecaBDE (often classified as BDE-209 [28]) and until 2015, hexabromocyclododecane (HBCD), which was withdrawn not because of toxicity levels [29], but more importantly, it had also been identified as a persistent organic pollutant or POP [30]. Resin formulations may be based on non-halogen or chlorine-containing acrylate-based or vinyl-based copolymers. Not only are these BrFR/ATO formulations effective flame retardants on all single fibre or multifibre-containing fabrics used in the UK upholstery market, but they do so while passing the durability requirement of a hot water-soaking treatment (i.e., flammability testing after soaking in water for 30 min at 40 °C) [27]. Furthermore, they have minimal effects on fabric aesthetics, which is important in furnishing fabrics where the surface structure, colour and handle of outer cover fabrics are of crucial importance.

Notwithstanding the success of the back-coatings and their effect in saving lives in UK dwelling fires [31], environmental pressures continue to mount against the use of all BrFRs in these and other textile applications. Before its effective ban in the EU in 2017, DecaBDE passed the 2004 EU risk assessment [32]. However, its manufacture and use were first banned in the USA at the end of 2013 at the same time it was included on the European Chemicals Agency list of Substances of Very High Concern (SVHC) and POPs and subsequently was banned from use within the EU [33].

### 2.1. Current Alternatives to HBCD and DecaBDE

Once the environmental consequences of using polybrominated aromatic materials, and especially DecaBDE used in textile back-coatings were under scrutiny, very similar alternatives such as 1,1′-(ethane-1,2-diyl)bis(pentabromobenzene) or 1,2-bis(pentabromophenyl) ethane (e.g., Saytex 8010, Albemarle, USA) were introduced as alternatives. However, while this molecule differed only from DecaBDE in the exchange of the ether –O- group by the ethane -CH_2_.CH_2_- group, and could effectively replace the latter in formulations quite easily, as early as 2007 the UK Environment Agency published a risk analysis [34]. Like DecaBDE, the flame retardant is very insoluble and at the time, while it recognised that it was not easily biodegradable and potential toxicological data with regard to aquatic species was not then available, its probable persistence in the environment was noted. At the present time it is under assessment by REACH in the EU as being potentially persistent, bioaccumulative and toxic. It is instructive to note that while there have been almost no academic publications examining its flame retardant properties, there are nearly 300 reporting its potential environmental hazards with its identification in the environment being reported as early as 2004 [35].

In answer to the persistence problems of monomolecular polybrominated aromatic flame retardants in general, industry has moved to adopting polymeric brominated species, which may be used as additives or as comonomers within coating and back-coating resins. Two common examples are poly(dibromostyrene), BrPS, and poly(pentabromobenzyl acrylate), BrPBz, which contain 59 wt% and 70 wt% bromine respectively [24], which are slightly lower values than for the above monomolecular BrFRs in which bromine levels > 80 wt%. Only the latter as 100% polymer or as copolymeric forms with other acrylic commoners are suitable for back-coating applications because of their compatibility with or copolymerisation with current typically used resins or resin monomers respectively. BrPBz especially was originally developed as a polymeric BrFR for use in engineering polymers during the 1980s, although its subsequent use in back-coatings has become widespread, especially as the concerns arising from monomolecular BrFRs increased [36]. Because BrPBz and copolymers are not covered by REACH regulations, they have not been subjected to risk analyses. However, since they are considered to be bonded within the coating, they are claimed to be less liable to enter the environment. Recent searches for the monomer pentabromobenzyl acrylate, as a possible degradation product of BrPBz in the environment have so far failed [37]. ICL Goup market flame retardant formulations based on BrPBz under their TexFRon^TM^ title (e.g., TexFRon^TM^P) and claim that this was the first bromine-based flame retardant to be recognized by Oekotex^®^.

### 2.2. Effectiveness of BrFRs in Textile Coatings and Back-Coatings

Surprisingly, even though there has been pressure to replace bromine in back-coating formulations for over 20 years, no publication has appeared that has actually demonstrated or benchmarked their effectiveness on a typical range of furnishing fabric compositions when, as fabric/PU foam composite, the face is subjected to an ignition source over a flammable filling such as flexible PU foam. In such a composite geometry the back-coating present on the reverse face must be able to transfer flame retardant activity to the otherwise flammable fibres on the front face of the fabric and act as a barrier in preventing ignition of the underlying, flammable, unmodified PU foam or other filling. Nearly all academic publications merely offer halogen-alternative treatments, usually for a single fibre such as cotton, without considering either the behaviour of the treated fabric as a barrier to ignition over a filling (as required in the UK regulations and as defined in BS5852: Part 1 (Source 1)) or that any such testing should occur after a 40 °C water soak pre-treatment [27]. These same authors appear also not to understand that if a BrFR replacement is to be developed what the magnitude of the real challenge entails. In the BS5852 standard, which all UK domestic furnishing fabrics should pass, the fabric face of a fabric/PU foam composite is subjected to a small butane flame (considered to a simulation of a “burning match”) for 20 s. After extinction of the igniting flame, the ignited fabric flame must self-extinguish within 30 s and show no traces of afterflame after 2 min.

Commercial experience since the adoption of the UK regulations about 30 years ago has shown that the following criteria, which BrFR/antimony III oxide (ATO) back-coating formulations fulfil, hold true:They function on any textile substrate, including the many fibre blends used in the industry, because of their predominant gas phase, flame quenching property and so are independent of the type of fibre present.The initial release of active Br^˙^ radicals and into the gas phase following thermal rupture of C–Br bonds within the flame retardant is simply represented by:
BrFR → Br^˙^ + FR^˙^(1)
Subsequent interactions with ATO produces volatile antimony-bromine species such as SbBr_3_ and antimony oxybromides [38] (see also Section 3.1) enable their diffusion to the front face of the fabric and extinguish any potential flame generation. In addition, these species may diffuse into an underlying substrate such as a polyurethane foam or other filling and extinguish any flames arising there. Thus the ATO synergized BrFR combination may be considered to provide both effective flame retardancy and the formation of a virtual barrier to flame penetration. Figure 1 presents a schematic representation of their mode of action.The presence of a char-forming resin component minimizes the effect of a thermoplastic fibre-containing face fabric, such as polyester, in that any hole size is minimized as well as allowing the released bromine radicals to quench any underlying PU foam ignition.

While the technology has been reviewed [25,39,40,41], the underlying scientific principles of back-coating have been little studied [42]. It was shown that for a typical DecaBDE/ATO formulation using a blade-over-air coating technique [42], the levels of penetration within a cellulosic upholstery fabric were dependent on blade angle, blade height and coat formulation viscosity. An optimum blade angle of 10° was observed, suggesting that in any back-coating process, selection of the blade variables is crucial if optimal coating and penetration are to be achieved without the coating reaching through to the front face (or creating so-called “grin-through”).The vapour or gas phase activity of the typical halogen-containing/antimony III oxide synergised flame retardant formulations [38] ensures that their activity may transfer easily from the coating on the rear face of the fabric to the front face where an igniting source such as a match or cigarette will impinge. Within the UK’s furnishing textile back-coatings market, a standard formulation based on antimony III oxide and brominated hydrocarbons, would typically comprise the total flame retardant (BrFR plus ATO): resin dry weight ratio = 1 with the relative amounts of BrFR and ATO in a mass ratio equivalent to a molar ratio Br:Sb = 3:1 reflecting the formation of the intermediate SbBr_3_ as the active flame retardant (see below). Such a formulation, made up as an approximate 50 wt% dispersion in water would be applied to the back of a cotton fabric at 20–30 wt% total solids add-on. For synthetic fibre-containing fabrics, back-coating levels are much greater because the char-forming character of the resin needs to offset the shrinking back and melting of the face fabrics, which would otherwise reveal the underlying PU filling to the igniting source. Back-coating levels here may be in the region of 50–100 wt%. Resins are most often based on vinyl or acrylic-based copolymers, usually including vinyl chloride or even vinylidene chloride as a comonomer [43]. The presence of chlorine adds to the overall flame retardancy of the resin, promotes char and also may act synergistically with bromine present in the BrFR [44].

Generally, non-halogen-based FR systems tend to function predominantly in the condensed phase, require more careful matching of flame retardant and fibre pyrolysis chemistries and so are therefore, fabric specific. In other words, a system that functions on pure cellulose may not function on other fibres, especially polyester and related blends, where melt dripping and hole formation expose underlying surfaces, usually a PU foam filling.

In attempts to demonstrate the effectiveness of BrFR/ATO back-coatings and based on the above cited work published 20 years ago [43], Eivazi working in our laboratories [45], determined the flammability properties of a number of typical examples as a means of comparing their behaviour with a series of sol–gel non-halogen back-coatings, which she was developing as potential replacements. Using a small scale test simulation of the BS5852: Part 1(Source 1) ignition condition (or simulated “match” test) based on one developed by the former company Mydrin Ltd., (now Lubrizol) over 20 years ago and described elsewhere [43], she examined the flammability behaviour of a number of currently available commercial back-coating formulations and compared them with a “standard” one comprising DecaBDE and ATO. Essentially, this test combines the sample dimensions of the BS5438: 1989: Test 2 vertical strip method with a 20 s flame application time as defined in BS5852: Part 1(Source 1). In this simulated test, a piece of non-flame retardant polyurethane foam of 220 × 150 × 22 mm^3^ (density of 22 kg/m^3^) is covered by a back-coated fabric sample with the foam adjacent to the back-coating. This composite is mounted in the Test 2 sample frame with the fabric face towards the gas burner in the horizontal mode (Test 2A, face ignition condition) with its tip 17 mm from the fabric surface (see also Figure 1). With a vertical flame height adjusted to 40 mm as specified in BS 5438, this is applied to the composite face for 20 s and then removed. If the composite continues to flame for more than 2 min or produce externally detectable amounts of smoke, heat or afterglow 2 min after removal of the ignition source, a “fail” is recorded for the test result, otherwise a “pass” result is reported.

Back-coated samples were also tested for their limiting oxygen index (LOI) values according to ASTM D2863. The materials and formulation components are listed in Table 1.

Proprietary formulations designated PolymBr and PolymBrP were supplied as dispersions in water of known solids and bromine content (see Table 1) and PolymBr/ATO and PolymBrP/ATO ratios were based on recommended proprietary recipes. The DecaBDE/ATO formulation was applied at a mass ratio equivalent to a bromine: antimony molar ratio = 3:1. Cotton fabric samples were coated at various dry back-coating add-on percentages, dried and cured. After water soaking each for 30 min at 40 °C (BS5651:1989) and then drying, samples were tested using the simulated “match” test.

In Figure 2 the LOI values for DecaBDE/ATO and PolymBrP/ATO formulations at respective simulated “match” test pass levels are almost the same (~25 vol%), although their respective bromine contents of 16.3% and 10.2% reflect the presence and effectiveness of the additional phosphorus-containing component with the latter formulation.

Thus it may be concluded that the presence of a phosphorus-containing species may reduce the level of BrFR required to generate a “match” test pass as was in fact demonstrated many years ago in our earlier research [43]. It is interesting, that the polymeric bromine-containing PolymBr/ATO formulation, while requiring the highest add-on (107%) to achieve a pass to the simulated “match” test is equivalent to a calculated bromine content 15.3%, just a little less than that for the DecaBDE/ATO coated fabrics, although a much higher LOI value of 31.7 vol% was determined.

Figure 3a shows images of cotton fabrics treated with each BrFR/ATO formulation after exposure to the igniting flame for 20 s at add-on percentages which yielded simulated “match” test “pass” conditions, i.e., where the afterflame and afterglow time ≤ 120 s. Figure 3b shows the associated respective fabric composite sample underlying PU foam damage. Table 2 shows the respective fabric damaged lengths and underlying PU foam damaged depths.

Optical microscopic images of fabric cross-sections of each back-coated fire-retardant fabric were investigated and the extent of coating penetration estimated and found not to be > 50% and so “grin-through” was not apparent. It was considered, therefore, that these tests demonstrated the effectiveness of organobromine-containing flame retardants in combination with antimony III oxide as back-coatings in terms of their ability when applied to a whole range of different fabrics to resist ignition by a simulated match source and provide a barrier to prevent ignition of the underlying and extremely flammable PU foam. Thus, when considering the development of halogen-free back-coating replacements, this is the level of performance to be achieved—a factor hardly ever considered in the numerous papers published in the academic literature.

### 2.3. Attempts to Replace BrFRs in Textile Back-Coatings

As stated above, since environmental concerns were raised about the use of halogen-containing flame retardant textiles, research focus has been in the main on reporting research into halogen-free flame retardants applied to cotton with blends and other fibres rarely being cited [39,46]. Furthermore, the need to consider durability to some washing or cleansing process has often been ignored and halogen-free back-coatings have received even less attention by the academic community. Historically while it was found 100% cotton fabrics treated with well-known durable flame retardant finishes such Proban^®^ and Pyrovatex^®^ and related chemistries [25] passed both BS5852: Part 1: Source 0 (cigarette) and 1 (simulated match) ignition conditions, soon after the UK regulations were enforced during the early 1990s, subsequently back-coatings comprising bromine-antimony-containing formulations were found to be not only be as effective and durable, but also they were much cheaper to apply. Furthermore, their application was not associated with potential losses in fabric strength and abrasion resistance.

The challenges posed by introducing non-halogen alternatives and their textile substrate specificity are exemplified by the limitations of phosphorus-based flame retardants, which are very effective on cellulosic- and wool-based fabrics, including blends with synthetic fibres only if the latter are in the minority. Furthermore, they are rarely efficient on 100% synthetic fibre-containing fabrics. Their general inability to transfer flame retardant activity from the back-coating applied to the rear of the fabric to the front face subjected to an igniting source is a prime reason for their failure (see Figure 1). For the last 15 years or so we have published results of our attempts to develop effective phosphorus-based back-coatings, principally on cotton, and while we have had limited success, a brief overview of this work will relate to the above more recent analysis of BrFR/ATO based back-coating performance. The strategies we have investigated range from a simple reduction in BrFR/ATO formulation levels to their elimination altogether.

#### 2.3.1. Reducing the BrFR Concentrations

Attempts to gradually reduce the DecaBDE/ATO content in a conventional formulation (comprising an acrylic, halogen-free binder) applied to cotton in combination with a number of bromine-free alternatives were initially investigated [43]. These latter included ammonium polyphosphate (APP), a cyclic oligomeric phosphonate, alumina trihydrate and zinc hydroxystannate and passes to the simulated “match” test described earlier were achieved at add-ons >30%, the level typically used for 100% cotton. When these non-halogenated FRs were applied individually, passes required much higher levels and in some cases were unachievable.

#### 2.3.2. Effectiveness of Phosphorus

Subsequent work [47] demonstrated the shortcomings of introducing simple char-forming, phosphorus-containing retardants (PFRs) into back-coating formulations applied to cotton, not least, their failure to pass the 40 °C water soak test. Here, selected PFRs, including some intumescents, formulated with selected resins, were applied as back-coatings to both cotton and cotton–polyester (35:65) blended fabrics. While all formulations, before water soaking, raised the limiting oxygen index, only those based on APP and a cyclic phosphonate enabled samples to pass the small-scale, simulated “match” test replicate for BS5852: 1979, Source 1. Back-coatings containing intumescents promoted higher levels of char formation but, while protecting the underlying PU foam, they were unable to transfer flame extinguishing activity to the front face of the fabric and thus extinguish it. While LOI is only a measure of fabric flammability with little indication of a potential fire barrier property, it was interesting to note that for both cotton and polyester-cotton fabrics back-coated with these PFRs, LOI versus fabric/PU foam composite test pass-fail relationships were observed where the fail-pass LOI region for cotton was over a fairly broad LOI range of 26.2–29.2 vol% compared with a much narrower range of 25.2–27.1 vol% for the polyester/cotton blended fabric.

Thermogravimetric analysis suggested that the more effective PFRs, as exemplified by APP, are those which liquefy by melting and/or decompose well below 300 °C, which enables wetting of the back face of the fabric and their diffusion to the front face where, as the temperature rises towards 300 °C, char formation occurs before ignition of surface fibres can take place. Thus, the liquefying property of APP and its diffusion across the fabric thickness replaced the diffusive property of Br˙ radicals and other volatile antimony-bromine-based species derived from BrFR/ATO formulations. However, all the applied PFRs had varying degrees of water solubility and so failed the 40 °C water soak, durability test demanded by the UK regulations [27].

#### 2.3.3. The Sensitisation of Decomposition or Flame Retarding Efficiency of Phosphorus-Based Systems

Based on the above observation and those of Lewin et al. [48,49] that small amounts of heavy metal ions can catalyse intumescence activity, attempts were made to reduce the melting/decomposition point of PFRs and hence increase their mobility in a back-coating geometry using this method. Inclusion of small amounts of certain transition metal salts, notably those of zinc II and manganese II reduced the onset of decomposition temperature of APP [50]. In the case of 2% manganese II sulphate addition, the onset temperature reduced from 304 °C to 283 °C and when applied in a back-coating formulation, LOI values increased slightly by about 1–1.5 units from 25.1 vol% for APP-only coated cotton to 26.6 vol% in the presence of 2% manganese acetate. However, the problem of durability to water-soaking still remained.

#### 2.3.4. The Introduction of a Volatile and Possible Vapour-Phase Active, Phosphorus-Containing Flame Retardants (PFRs)

It was reported by Hastie and Bonnell over 35 years ago, that volatile P-containing radicals are as efficient as Br˙ in quenching flame propagation reactions [51]. Four potentially volatile PFRs were selected based on their reported boiling or decomposition data [52]. TGA studies of monomeric cyclic oligomeric phosphonate (Antiblaze CU, Solvay), tributyl phosphate (TBP), triphenyl phosphate (TPP) and triphenylphosphine oxide (TPPO) suggested that TBP (b.pt. = 289 °C with decomposition) would be most suitable for furnishing fabrics because its initial decomposition temperature (150°C), is well below the melting temperature of polypropylene (~165 °C) and the ignition temperature of cotton (~350 °C), both which are commonly used fibres in 100% compositions in furnishing fabrics. TBP was combined with the intumescent char-forming agent, Great Lakes NH 1197(formerly Chemtura, now Lanxess) comprising phosphorylated pentaerythritol. When back-coated on to 220 g/m^2^ cotton and 260 g/m^2^ polypropylene fabrics respectively to achieve nominal dry add-ons in the 40–70% range, the best results were obtained both for increased LOI and tendency to pass the simulated “match” test with the more volatile TBP in a mass ratio, intumescent: TBP = 4:1. Further evidence of the volatile phosphorus activity was gained by determining the retention of phosphorus in charred residues from back-coated samples containing APP, melamine phosphate (MP), Antiblaze CU, and the oligomeric phosphate–phosphonate Fyrol 51 (formerly Akzo, now ICL IP, withdrawn) where the two last are liquids and are potentially vapour-phase active flame retardants. Phosphorus loss was lowest for fabrics containing the char-promoting APP and MP and highest for Antiblaze CU and Fyrol 51, which also exhibited the highest coated fabric LOI values. These results suggested that an ideal back-coating might comprise a non-volatile, char-former like APP or MP in combination with volatile phosphorus- or even nitrogen-containing species. Introducing melamine (Mel) raised the LOI values of all samples above 27 vol%, which also passed the simulated “match” test before water-soaking. Again, however, water-soak resistance was lacking, although an APP/Mel/Fyrol 51 formulation after soaking yielded a “match” test pass after a 10 s ignition time.

Clearly, the replacement of BrFR/ATO formulations remains a challenge, where in back-coated textiles and similar applications in particular degree of flame retardant transferability is required coupled with acceptable levels of water-soaking and/or wash durability.

It is interesting to note, that in spite of the many papers published recently in which applications of nanotechnology with the aim of improving non-halogen-containing, flame retardant efficiency [53] coupled with the interest in novel surface treatments mainly on cotton, such as sol–gel [39,54,55,56,57], layer-by-layer [39,54,57,58,59,60,61,62,63,64,65] and atmospheric plasma [66,67,68], limited levels of flame retardancy and durability have often only been achieved. None of these treatments to the author’s knowledge has been either applied to or found to be acceptable in back-coating applications.

### 2.4. “Biobromine” as an Environmentally Acceptable Source of BrFRs

The question might be asked, “Is it the element bromine or brominated flame retardants in general that fire the current debate?” To some environmentalists, the answer will be both, but then bromine is not an unusual element in the biosphere. Bromine occurs in the environment primarily as the bromide ion and occurs, not surprisingly together with chloride-containing minerals as well, in sea water where it typically comprises about 65ppm, although in the Dead Sea it is at a level of about 5000 ppm. Not surprisingly perhaps, marine and subsequently terrestrial life can accommodate these levels and has, over time, biosynthesised an armoury of naturally-occurring, organobromine or “biobromine” products. These are included within the >3800 known organohalogen species, which are reported to comprise about 2200 organochlorine, 1950 organobromine, 95 organoiodine and 100 organofluorine compounds [69]. This high level of organobromine species may appear to be surprising since the [Cl^−^]/[Br^−^] mole ratio in the marine environment is of the order of 500:1. However, this is considered to be a consequence of the lower oxidation potential of the bromide ion to a bromine atom reaction with respect to chlorine, which can increase the likelihood of bromide ions entering bio-organic synthetic reactions [70]:Cl^−^ − e → Cl   −1.36 V(2)
Br^−^ − e → Br   −1.07 V(3)

These values reflect the respective C–Cl and C–Br bond average energies of 339 and 276 kJ/mol. However, actual bond energies in a given molecule are determined by the structural features present. Most organobromine flame retardants are based on aromatic structures, hexabromocyclohexadodecane (HBCD) being a notable exception, since the aromatic C–Br bonds present are more stable than aliphatic C–Br bonds, which enable aromatic BrFRs to withstand higher processing temperatures, while functioning effectively as retardants. However, the strengths of C–Br bonds are very dependent on the aromatic environment. For example, while bromobenzene is reported to have a bond dissociation energy of 296 kJ/mol, which is greater than the value in methyl bromide of 280 kJ/mol, aromatic C–Br bond dissociation energies are reduced by the so-called “ortho effect” where two aromatic rings are fused together in the ortho- position, thus demonstrating the effect of molecular environment [71]. Recent work by Kazakbayeva et al. [72] has corroborated this earlier observation and shown significant variance of C–Br dissociation energies between more complex bromoaromatic compounds.

It has been reported that a major issue with DecaBDE is its ability to be degraded by UV to the known toxic penta- and octa-congeners [8,14,15], which again suggests that there are differences of C–Br bond energies within the same molecule. Obviously if BrFRs are to be able to be biologically degraded to non-toxic products, the means of either weakening C–Br bonds and /or rendering them more amenable to enzymatic reduction to bromide ions should be better understood. Conversely, it has been suggested that species such as Br^+^ might be involved and it is known that some bromoperoxidase enzymes can catalyse the oxidation of Br^−^ ion by hydrogen peroxide to hypobromous acid, which can then brominate an organic species [73].

#### 2.4.1. Examples of Naturally Occurring Polybrominated Aromatic Compounds

Many of these naturally occurring organobromine compounds are polybrominated as reviewed by Faulkner over 20 years ago [70,74]. For instance, marine bacteria, green and red algae and sponges are particular producers of brominated metabolites, which generally possess antimicrobial activity used to the advantage of the respective hosts. Examples of polybrominated species from green and red algae and sponges are shown respectively in Figure 4a–c with a preponderance of brominated diphenyl ether derivatives illustrated. Figure 4c is of particular interest and these examples are described as minor metabolites of *Dysidea herbacea* [74,75,76,77]. The BDE derivative metabolites in Figure 4b and especially the pentabromo-substituted examples in Figure 4c, bear striking resemblances to current BrFR structures.

It is evident that unless the oceans are accumulating these polybrominated species, there must be as yet unknown biodegradative paths, which convert the organobromine present back to bromide ions. It is thus most likely that there are a large number of “bromine cycles” in existence, which if more understood, would enable the development of a group of so-called “biobromine”-based flame retardants (BioBrFRs) that could be accommodated within these natural cycles. Faulkner [74] has also proposed that biosynthesis and biodegradability could also be related to those bromine-containing compounds that are chiral, although the examples above are excluded from this proposal.

While the fate of such naturally occurring organobromine compounds in the marine environment is not well understood, there does seem to be an association between bromine and organic carbon in marine sediments with evidence of biogeochemical cycling of bromine between organic and inorganic (notably as Br^−^) forms reducing in concentration with increasing depth [78]. Such cycling may include debromination by microbial activity and very recent research has suggested that, while the maximum amount of energy that is potentially provided through organobromine respiration is low relative to other metabolic pathways, such as sulphate reduction and methanogenesis, it may still be significant [79]. Furthermore, these same workers report that dissolved Br^−^ and total solid-phase bromine concentration profiles indicate that the most rapid debromination occurs in the upper 10–20 m of a typical sediment column.

Therefore, the exemplar, naturally occurring polybrominated structures in Figure 4 must be a part of one or more organobromine-to-bromide cycles, which suggests that there may exist novel, biocompatible BrFR solutions, if they can be shown to be compatible with and accommodated by such a cycle. However, because some of these species have antimicrobial and possibly toxic properties, which enable them to be used for defence against predators [70,74], they could perhaps be more safely used if incorporated physically or chemically into polymeric structures.

#### 2.4.2. Factors Influencing Biodegradability of Brominated Flame Retardants

Xiong et al. [8] have recently reviewed the whole issue regarding the fate of a number of brominated flame retardants in the environment including literature regarding their exposure in both abiotic, including air, water, dust, soil, sediment and sludge, and biotic matrices, including bird, fish, and human serum. Furthermore, they review the effects of photodegradation, thermal degradation and biodegradation. Once in the environment, apart from bioaccumulation, significant concerns have been specifically raised about the effect of UV light and its ability to convert agents like DecaBDE, which have passed risk assessments regarding their general toxicity into more toxic, debrominated congeners like octa- and pent- derivatives [80]. Photo-debromination and the related kinetics have been shown to be a stepwise process [14], which appears not to be easily explained in terms of relative orientations of bromine presence in ortho-, meta- and para-positions within respective aromatic rings [15]. Very recent work suggests that ease of photo-debromination reactions in polybrominated aromatic compounds substituted with the highest number of bromine atoms is defined by the largest stretching of the C–Br bond in the first excited state with theory indicating that that, excitations of BFRs proceed via π→π*, or π→σ* or n→σ* electronic transitions [81]. Not least is the effect of a fully brominated aromatic ring, which will create a steric hindering effect between adjacent bromine atoms, thereby encouraging the first stages of debromination. While BrFRs like DecaBDE have received considerable attention, UV degradation of decabromodiphenyl ethane or 1,2-bis(pentabromophenyl) ethane has received much less and Najia et al. [16] have shown it can photolytically undergo an efficient stepwise reductive debromination that follows first-order kinetics. With regard to the polymeric brominated flame retardants, even less is known and brominated polystyrene, for instance, although not used in textile back-coatings, UV irradiation yields 75 different degradation products, while thermal degradation led to a significantly lower number [82]. These same authors claim that possible toxicological concerns regarding this polymeric BrFR had not yet (2019) been the subject of peer-reviewed publications [83].

That two of the most previously, commonly used BrFRs, for textile back-coatings were decabromodiphenyl ether, DecaBDE (also often referred to as BDE 209), and hexabromocyclododecane, HBCD, were identified as “persistent organic pollutants” or POPs and “substances of very high concern” or SVHC [30,33], reflects not only their stability in the environment, but also their potential toxicological properties, especially with regard to their respect products of biodegradation. Waaijers and Parsons have recently reviewed this whole area as it relates to BrFRs in general [18] in which they specifically cite the works of Vonderheide et al. [84] and Gerecke et al. [85] in regard to the anaerobic debromination of a number of BrFRs, including BDEs and HBCD. More specifically DecaBDE, was observed to degrade to the toxic metabolites octa- and pentabromodiphenyl ethers, in sewage sludge for periods up to two years and both HBCD and DecaBDE degradations followed pseudo first order reactions with rate constants in the order HBCD >> DecaBDE [85]. These rates were independent of other nutrients and DecaBDE exhibited a half-life of 700 days. Clearly, debromination reactions are of particular significance in the biodegradation of both DecaBDE and HBCD and most likely figure significantly in both anaerobic and aerobic conditions. However, besides differences in respective C–Br bond strengths, the nature of the debromination reactions will be generally different. This is because removal of bromine from a polyaromatic structure like DecaBDE will be essentially a reductive one to free Br^−^ ions, whereas that for the aliphatic HBCD will be dehydrobromination leading to unsaturated cyclododecene derivatives. Technical grade hexabromocyclododecane comprises three chiral diastereomers, α-, β- and γ-HBCD, each existing as pairs of enantiomers [29], which will influence the degradative pathways. Investigations by Davies et al. [86] of the biodegradation of technical HBCD mixtures in soils and freshwater sediments showed that for γ-HBCD, surprisingly no brominated degradation products were identified and that no biodegradation was observed for the minor α- and β- diastereomers. Further studies with wastewater sludge and sediments identified of tetrabromocyclododecene, dibromocyclododecadiene and cyclododecatriene as metabolites from all three diastereoisomers [87]. However, a very recent study by Chang et al. [88] noted that optimum HBCD biodegradation occurred by *Rhodopseudomonas palustris* strains isolated from paddy field soil occurred at neutral pH and at 35 °C. Relative molar ratios of the released bromide ions were observed in the range between 1 and 3.5, suggesting that the debromination reactions occurred, with the concurrent production of two metabolites, pentabromocyclododecanol and pentabromocyclododecene. The presence of metal ions also had a significant effect on HBCD biodegradation with iron II sulphide, which is often in anaerobic sediments, in particular having been shown by Li et al. [89] to accelerate the reductive, sequential dibromoelimination of HBCD to form 1,5,9-cyclododecatriene, suggesting its use as a reactive agent for treating HBCD-contaminated sediments.

However, recent research has also indicated that certain BrFRs and especially polybrominated diphenyl ethers transform to hydroxylated and methoxylated derivatives, which while raising concerns about their potential ecotoxicity [90], have been shown to be able to be metabolised under certain conditions [91]. This suggests that the introduction of potentially bioactive functional groups into the BrFR structures, will improve their ease of biodegradability. This would not be too surprising, given that the naturally occurring polybrominated aromatic species exemplified in Figure 4, have either –OH or –OMe functional groups present. This then raises the possibility of correctly functionalised synthetic BrFRs entering into existing geobromine cycles in both marine [72,77] and even terrestrial [92] environments.

## 3. Organobromine Flame Retardant Synergists

Given the above-mentioned toxicological concerns raised regarding all BrFRs and their use, which now include the synergist antimony III oxide [22,23], then a brief overview of how it functions, both in terms of its advantageous and disadvantageous characteristics, is instructive in order to inform whether alternatives available now should be more effectively considered.

### 3.1. Antimony III Oxide

While it has been recognised that a number of antimony compounds act as synergists for halogen-containing flame retardants, antimony III oxide, Sb_2_O_3_, is that most commonly used, largely based on its lower cost and its apparent properties of being independent of BrFR compound chemical structure and polymer type in which it is present. The history of its use as a potential flame retardant in its own right goes back over a century [93], although its use as a synergist in combination with chlorine-containing species stems from World War II and work by the US Army in 1944, as described fully by Little later in 1947 in what must have been one of the first texts describing the flame retarding of textile fabrics [94]. This early work identified the role of hydrogen chloride as being of major importance. At the same time, Coppick of the American Viscose Corporation [95] identified a number of important features regarding the effect of ATO or Sb_2_O_3_ in the presence of a neoprene rubber coated on to cotton for use as military tentage including:the effectiveness of the Sb_2_O_3_/neoprene combination in regard to HCl generation;the identification of an optimum Sb_2_O_3_/Cl ratio equivalent to antimony oxyhalide, SbOCl, formation as an intermediate;the use of additional boric acid, zinc borate or an ammonium phosphate as afterglow retardants; andthe combination of the neoprene/Sb_2_O_3_ combination alone was wash durable, but not afterglow retardant.

Since that time a significant amount of work has been undertaken to understand the mechanism of chlorine-containing flame retardant-Sb_2_O_3_ synergism, notably by Costa et al. [96,97], which has identified a series of reactions chlorination reactions of the synergist via a series of complex volatile antimony oxyhalides leading eventually to the formation of SbOCl and then SbCl_3_. Parallel work showed that for DecaBDE-Sb_2_O_3_ combinations, a similar sequence of reactions occurred with the generally more rapid formation of SbBr_3_ [98]. This most likely explains why in antimony-bromine flame retardants generally, including those used in coating and back-coating applications [25], the optimum antimony: bromine molar ratio = 1:3. The main flame retarding activity of both Sb-Cl and Sb-Br formulations involves reaction or either SbCl_3_ or SbBr_3_ with the major flame propagating radicals H·, O·, OH·, and HO_2_· and derived hydrogen halides, HX, via a series of complex chain reactions [38,99]. The principal propagation and termination reactions may be more simply summarised as:SbX_3_ + H· → HX + SbX_2_·(4)
SbX_2_· + H· → SbX· + HX(5)
SbX· + H· → Sb + HX(6)
HX + H· → H_2_ + X· (leading to flame inhibition)(7)
X· + HO_2_· → HX + O_2_(8)
HX + OH· → H_2_O + X·(9)

Hastie also proposed other termination/flame inhibition reactions based on reactions of the type [99]:SbOH + H· → SbO + H_2_(10)
SbO + H· → SbOH(11)
where the formation of antimony oxy species derives from reaction with atomic oxygen O· or OH· radicals. These stages are similar to those proposed also for the flame extinguishing effects of tin compounds as shall be discussed in the next section.

Generally, Sb_2_O_3_ or ATO appears to be equally efficient with the current commercial range of chlorinated and brominated flame retardants available, thus suggesting that the above chemical mechanisms are little affected by the chemical character of the halogenated flame retardant component or the polymer matrix they are present in or textile material they are applied to. Furthermore, while the gas phase mechanisms as outlined above are considered to be the major, if not sole flame retardant mechanism operating in a given polymer matrix, since there is evolution of either HCl or HBr, these as acids, will promote condensed phase char-forming reactions in polymers like cellulose, poly(vinyl alcohol) and poly(vinyl acetate) (and related copolymers), via dehydration of the pendant –OH groups present. In coating and back-coating formulations, this char-forming character of the resin component present is crucial in offering an additional char barrier to the exposed textile substrate. This is particularly important if the underlying textile is simply fusible and not char-forming like polypropylene and polyester, commonly used in both domestic and contract furnishing fabrics. In poly(vinyl chloride)-based coatings for textiles, the need for the presence of a plasticiser, which reduces the inherent flame retardant effect of the chlorine content, requires the presence of ATO to restore it to an acceptable level. Because of its very low particle size (see Table 3), antimony III oxide may be easily extruded as a dispersed additive in fibres and so when included in modacrylic fibres, which typically comprise a copolymer based on acrylonitrile and vinylidene chloride, their inherent flame retardancy is enhanced, although the presence of these fine particles may change the fibre lustre and aesthetics of the resulting fabrics.

However, because of the very effective gas phase flame inhibiting reactions of these systems exemplified in Equations (7), (10) and (11) above, incomplete combustion obviously occurs with increased concentrations of toxic fire gases [100] and especially smoke [101] being the outcomes, both of which are the major causes of loss of life in fires. In addition to the above flame inhibiting reactions, the concentration of complex particulate materials, including complex polynuclear aromatic hydrocarbons [102], will be enhanced by the debromination and dehydrobromination of the BrFRs present. Thus antimony-bromine flame retardant formulations have significantly increased smoke and carbon monoxide levels associated with them relative to the pure polymer matrix during burning.

Furthermore, afterglow may also be a problem as first documented many years ago, as was its suppression by the addition of ammonium phosphate and zinc borate [95]. Zinc borate not only is a smoke suppressant but also interacts positively with antimony III oxide in its catalytic activity [103]. However, while the toxicity of zinc borate did not cause concern 20 years ago [104], more recently it has been classed as a Category 2 Reproductive Toxicant according to the Global Harmonized System (GHS) for Classification and Labelling of Chemicals and is toxic to aquatic life [105]. Not surprisingly, these factors have added to the ecotoxicological concerns regarding the use of halogen-containing flame retardants in recent years.

While a possible ban or restriction of use of antimony III oxide and other antimony-based synergists may be called for in the future, there are conflicting reports with regard to its commercial importance in any case. For example, market predictions suggest that at the present time about 50% of antimony used in the world is as synergists for halogen-containing polymers and flame retardants with the likelihood of continued growth in total global usage [106]. This also suggests a continuing increase in the use of halogen-containing flame retardants and polymers in spite of increasing environmental pressures. On the other hand and in spite of the previously mentioned toxicological concerns, there have been pessimistic predictions about its continuing availability as a globally sourced mineral to the extent that unless serious recycling of antimony III oxide is undertaken, the world’s known reserves will have become sufficiently depleted to be uneconomically viable before 2050 [107].

### 3.2. Zinc Stannates

At the present time, the only commercial alternatives to antimony III oxide as an effective synergist are zinc hydroxystannate (ZnHS) and zinc stannate (ZnS) and their development since their origins in about 1990 have been reviewed [108]. Their general characteristics are also listed in Table 3. They are generally assumed to behave in a manner similar to antimony III oxide in their synergistic activity with halogenated polymers and flame retardants but in addition, they have considerable smoke and carbon monoxide suppressing activity [109] and may also promote char formation. Both ZnHS and ZnS are used successfully in applications involving key polymers like PVC coatings, polyamide engineering plastics and unsaturated polyester resins used in combination with fibre and textile reinforcing elements. However, unlike antimony III oxide, both ZnHS and ZnS are genuine mixed oxides where the zinc and tin atoms are built into a crystal lattice rather than as simple oxide blends. It is the chemical arrangement of the zinc and the tin within the crystal structure which gives these materials their fire protection performance in combination with halogen-containing species, although differing is some respects from the chemically simpler ATO in terms of selectivity of synergistic efficiency with different BrFRs and polymer or textile substrates (see below).

Whether or not ZnHS or ZnS is chosen for a particular application depends on the processing temperatures used, since in the former its hydroxyl groups are driven off by heat at about 180°C in the initial stages of a fire, thereby giving additional cooling and slowing the combustion reaction. Zinc stannate, however, is stable up to 400 °C and so can withstand processing in melt extrusion and similar processes in polymers that may melt at temperatures approaching 300 °C. A major feature is that at the present time, both stannates have had no undesirable toxicological properties identified and so they are considered to be more environmentally sustainable than ATO. Furthermore, unlike the latter, which shows no flame retardant activity when used alone unless present in a halogenated polymer like PVC, both ZnHS and ZnS can be used alone in non-halogenated polymeric systems as char-promoters and smoke suppressants [108].

In fibre-forming polymers like polypropylene [110] and polyamides 6 and 6.6 [111] synergistic behaviour of the zinc stannates with halogenated additives has been demonstrated, although because of the relatively high levels of flame retardant required, acceptable fibres have yet to be reported. Unfortunately, very recent research by Eivazi [45] has shown that the replacement of ATO by ZnHS in two almost identical formulations containing either DecaBDE or a polymeric BrFR applied as a back-coating to cotton fabric at similar respective add-ons, failed the small-scale ignition simulation test of BS5852 when tested over unmodified PU foam, unlike the respective antimony-containing formulations. These rather disappointing results further illustrate that unlike Sb-Br containing formulations, the effectiveness of zinc stannate-containing analogues is dependent on the chemistry of the BrFR and the polymeric substrate, whether as a bulk polymer or applied to a textile. This is perhaps not too surprising since the apparent mixed oxide character of both ZnHS and ZnS suggests that each oxide component (ZnO or SnO_2_) might interact differently with a given BrFR and in the presence of a different polymer.

Initial mechanistic work by Cusack et al. [112] proposed that an ideal mole ratio Sn:halogen = 1:4 is considered to be the optimal ratio based on the observed volatility of tin or even 6:1, if zinc is included, with the intermediates SnX_4_ and ZnX_2_ being analogues to SbX_3_ intermediate formation observed with antimony-halogen systems. However, while vapour or gas phase activity of zinc stannate-halogen systems is considered to be the more significant flame retardant mechanism, chars may still contain considerable fractions of tin and zinc [113,114]. Increased char formation has been observed in parallel with smoke suppression [115].

As stated above, Hastie as observed for antimony synergised systems [99], had earlier proposed that tin compounds functioned via H· radical interactions of the type:SnOH + H· → SnO + H_2_(12)
SnO + H· → SnOH(13)
SnO_2_ + H_2_ → SnO + H_2_O (14)
which were later corroborated by Cusack et al. [116]. However, later work by Kicko-Walczak [117,118] reported the mechanistic studies of action of zinc hydroxyl stannate in brominated unsaturated polyester resin matrices as being a multi-stage degradation giving rise to the initial formation of tin II and IV bromides over the range 240–340 °C.

At higher temperatures (340–420 °C), the final char structure is formed and the vapour phase flame retardant reactions take place. The released tin bromides are considered to be hydrolysed in the flame to tin II oxide and hydrogen bromide, with both products then inhibiting flame reactions as outlined above and as formerly proposed by Hastie [99]:SnBr_2_ + H_2_O → SnO + 2HBr(15)

The above suggested mechanisms, coupled with their inherent, little understood smoke suppression activity, clearly indicate that zinc stannate-halogen flame retardant interactions are not as straightforward as reported for Sb-Br systems. In this respect, it also is noteworthy that recent work in our laboratories [119] has reported that zinc stannate in combination with poly(pentabromobenzyl acrylate), BrPBz, in polyamide 6.6 does not show the above expected vapour phase activity via volatile tin II and IV bromides, tin (II) oxide and interactions between the latter and released hydrogen radicals as being the most significant mechanism. In fact a considerable amount of bromine is trapped within the char that would otherwise be expected to have been released into the vapour phase and in addition the interaction of bromine is primarily with zinc present in ZnS and not with tin, as might be expected from earlier studies [112,114,118].

### 3.3. Metal Tungstates

With no real advances during the last 30 years towards seeking a replacement for antimony III oxide apart from the zinc stannates, recent research by ourselves has investigated over 150 metal complexes for their ability either to promote char and/or demonstrate synergistic activity with brominated flame retardants in an engineering polymer typified by polyamide 6.6 (PA66) where BrFRs are often preferred [120] and the application can withstand the added cost of an ATO replacement. Initial studies with zinc oxalate in combination with BrPBz showed positive, possibly synergistic interactions in terms of reduction in cone calorimetric peak heat release rate (PHRR) and increased residual char levels [121]. Subsequent work showed that aluminium (AlW), tin (II) (SnW) and zinc (ZnW) tungstates not only increased char formation and reduced PHRR values when present alone in PA66 [122], but when also in the presence of the phosphorus-containing FRs, aluminium diethyl phosphinate (AlPi) and AlPi in the presence of melamine polyphosphate, they increased their respective flame retardant behaviours and, in the case of ZnW, reduced smoke formation [123]. This work was subsequently extended to study potential synergistic interactions with the BrFRs, brominated polystyrene and poly(pentabromobenzyl acrylate) [124].

Each tungstate was compounded alone (at 5 wt%) and with either of the two BrFRs, BrPS and BrPBz (at 10 wt% bromine levels) in PA66 and compared for fire performance with each BrFR alone, present also at 10 wt% Br levels. These results are summarised with respect to limiting oxygen index, LOI, and cone calorimetric peak heat release rate, PHRR, percentage reduction in PHRR, RPHRR, and total smoke smoke release, TSR, values in Table 4.

It is evident that the addition of the three tungstates alone has little effect on LOI values, although reductions in PHRR values are noted with respect to the PA66 control. However, when tin II and zinc tungstates were added to the respective BrFR-containing formulations in PA66, there were significant increases in LOI indicating possible synergy. The ZnW-BrPBz formulation achieved the highest LOI value of 28.5 vol% with aluminium tungstate showing a minimal effect. However, the post-ignition parameter, PHRR, shows signification reductions for both SnW- and ZnW-BrFR formulations, with the ZnW-BrPS showing the highest percentage reduction (R_PHRR_ = 70.5%). As expected, significant increases in TSR values occur when each BrFR is present alone in PA66, but considerable reductions with respect to these values are observed for both zinc tungstate-BrFR formulations.

In previous work [125], the relative flammability properties of BrPS and BrPBz formulations containing either antimony III oxide or zinc stannate, ZnS, in PA66 were compared and these results for total smoke release have been combined with those in Table 4 to produce the differential smoke (ΔTSR%) bar charts in Figure 5 [124]. ΔTSR% represents the percentage changes in smoke release caused by the addition of ATO, ZnS or tin II and zinc tungsates to each BrFR-containing formulation in PA66.

Not surprisingly, the formulations containing ATO as the synergist show significant increases in smoke generation with only SnW behaving in a similar but much less severe manner. In contrast, the well-documented smoke suppressing property of zinc stannate, in the presence of each PolyBrFR is evident [107]. However, the greatest reduction in TSR is observed when ZnW is in the presence of either BrPS or BrPBz, which suggests that ZnW is comparable to zinc stannate as a smoke suppressant. It is interesting to note that the effectiveness of ZnW compared to ZnS with the two BrFRs investigated is reversed, in that ZnW is the more effective smoke suppressant with BrPS, while ZnS is the more effective with BrPBz.

Analysis for metallic and bromine residues in cone calorimetric chars [124] showed that generally there was a loss of bromine to the vapour phase as expected, as indicated by a clear reduction in the Br:W molar ratios present. However, while losses of Zn and Sn were also observed, surprisingly the former was significantly greater. Furthermore, higher Br:W ratios were observed for the ZnW-containing samples, which would suggest that more bromine was retained in the condensed phase than with SnW-containing formulations. These char results indicated that respective metal/bromine losses to the volatile phase are not simply comparable for the two tungstates and that condensed phase activity of ZnW is significant in its role as both a synergist and a smoke suppressant, especially with brominated polystyrene.

If metal tungstates and in particular zinc tungstate are to be considered as possible replacements to antimony III oxide, it is essential that their properties in combination with a larger range of BrFRs present in a wider range of polymers and/or applied to textile fabrics of various types should be studied. Given that the zinc stannates are very selective in regard to their synergistic behaviour, it would not be surprising if zinc tungstate behaved in a similar manner based on the above results. Of more importance to this review, is its possible role as an ATO replacement in textile coating and back-coating applications as well as in fibres such as modacrylics.

## 4. Conclusions

In a very recent review [126], the authors state that developments in back-coatings (and coatings) for textiles have not occurred because of the inability to overcome the removal of halogens, as we have noted previously [43,47,50,52]. This present review, however, has pointed out that the removal of halogens may not need to be an essential requirement to alleviate the real and often perceived environmental concerns regarding brominated flame retardants in current use. The challenge in replacing currently available BrFRs successfully for the whole range of polymers and textiles serviced by them, possibly lies in synthesising alternative structures that mimic naturally-occurring organobromine-containing species. Of the almost 2000 of the latter, which have been identified in both terrestrial and marine environments, of which a significant number are functionalised and polybrominated in character, including diphenyl ether derivatives, there is the need to assess them for their potential flame retardant properties. Since these naturally occurring polybrominated structures are a part of defined biosynthetic-biodegradative pathways, there is also the need to more fully understand these and then observe whether synthetic analogues fit within these naturally existing biodegradative cycles.

In parallel, while the conventionally used synergist antimony III oxide functions very effectively independently of both BrFR chemical structure and the polymer matrix or textile substrate present, its replacement by the established non-toxic, but more expensive zinc stannates with naturally-occurring polybrominated species also requires investigation. However, given their greater selectivity in their ability to be effective as BrFR synergists, combinations with selected BioBrFRs would need to be applied to a variety of polymer and textile structures so that optimal synergist-bioBrFR choices may be identified. In a similar manner, the recently reported synergistic and smoke suppressing effects of zinc tungstate in combination with selected polymeric brominated flame retardants compounded in polyamide 6.6 also offers opportunities for ATO replacement in the longer term.

In conclusion, it is proposed that the future of sustainable BrFR innovations should be based on replicating naturally-occurring, polybrominated structures or BioBrFRs developed in conjunction with non-toxic, smoke suppressing synergists such as the zinc stannates or zinc tungstate, carefully tailored for given polymeric and textile substrates.

## Figures and Tables

**Figure 1 polymers-12-02160-f001:**
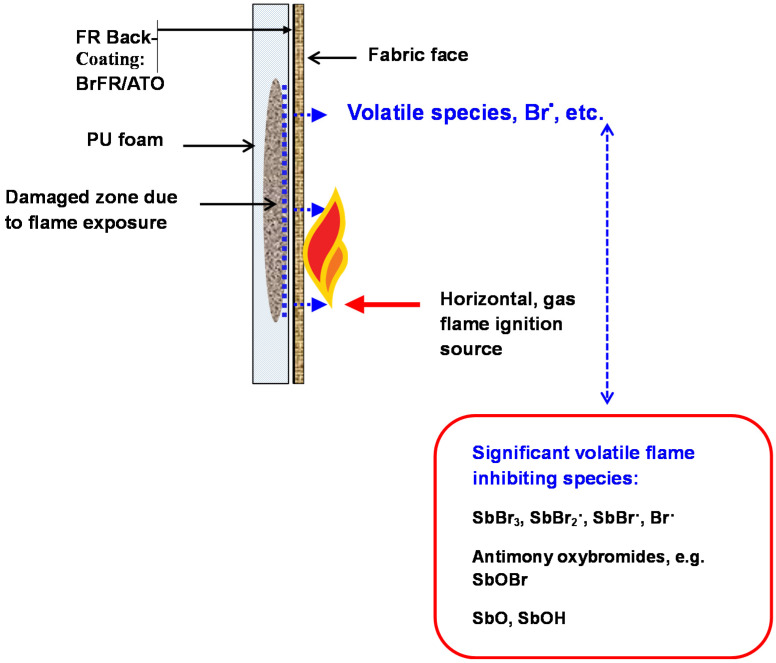
Schematic diagram of small flame ignition of a brominated flame retardant/antimony III oxide, back-coated fabric-covered PU foam to illustrate bromine radical formation and diffusion (adapted from reference [39]).

**Figure 2 polymers-12-02160-f002:**
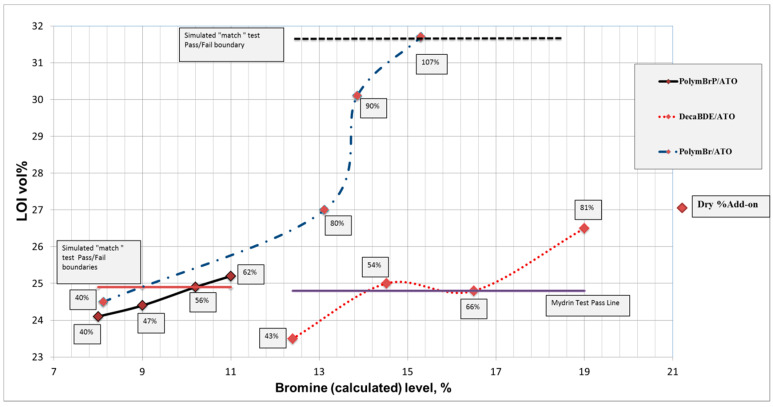
Effect of Bromine content on LOI values of DecaBDE/ATO, PolymBr/ATO and PolymBrP/ATO back-coated cotton fabrics with “match” test “pass/fail” boundaries indicated [45].

**Figure 3 polymers-12-02160-f003:**
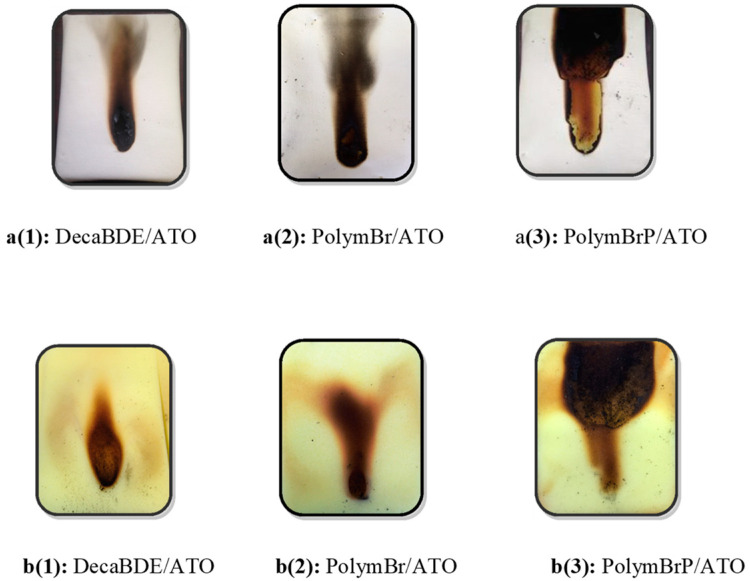
Simulated “match” samples at the “pass” condition; (**a**) fabric and (**b**) foam damaged areas [45].

**Figure 4 polymers-12-02160-f004:**
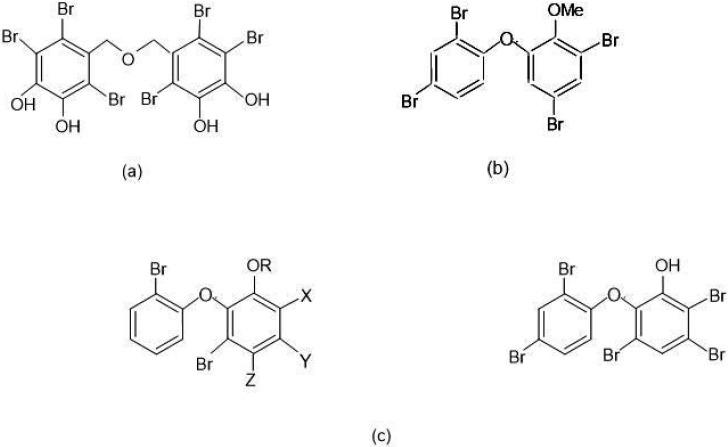
Polybrominated metabolites from marine (**a**) Red algae, (**b**) Green algae and (**c**) sponges, where (**i**) R=X=H; Y=Z=Br (**ii**) R=Z=H; X=Y=Br (**iii**) R=Y=H; X=Z=Br (**iv**) R=Me; Y=H; X-Z=Br [70,74,75,76,77].

**Figure 5 polymers-12-02160-f005:**
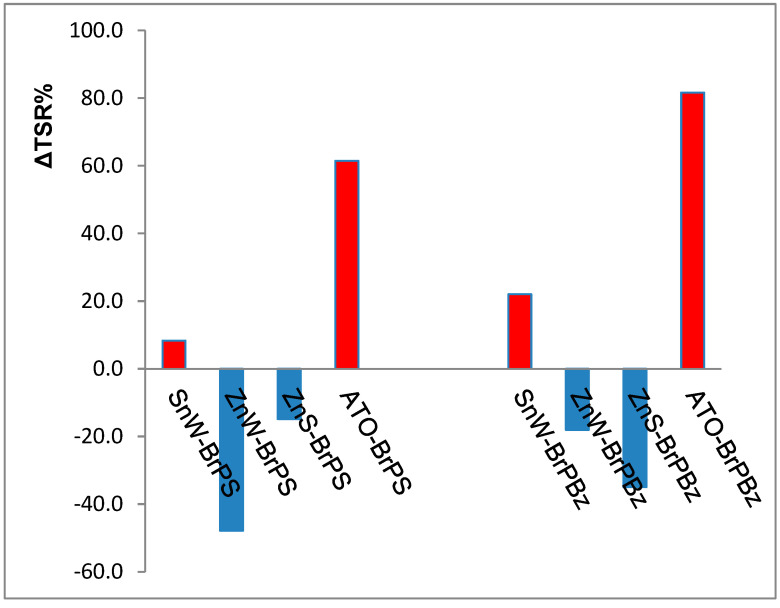
Percentage changes in smoke generation where the percentage change in total smoke release, ΔTSR% = (TSR_(MC/BrFR)_/TSR_(BrFR)_ − 1) × 100 and TSR_(MC/BrFR)_ and TSR_(BrFR_ are the respective smoke release values for each synergist and tungstate (MC) in combination with each BrFR and each BrFR alone in PA66: Red columns indicate an increase and blue columns a decrease in smoke generation with respect to that from either brominated polystyrene, BrPS, or poly(pentabromobenzyl acrylate), BrPBz, present in PA66 alone. SnW and ZnW = tin (II) and zinc tungstates respectively; ATO = antimony (III) oxide and ZnS = zinc stannate [124].

**Table 1 polymers-12-02160-t001:** Materials used for back-coating cotton with BrFR/ATO formulations [45].

Material	Commercial Name and Supplier
Acrylic co-polymer resin	Hycar T-91 (Lubrizol)
BrFR flame retardants:	Decabromodiphenyl Ether, DecaBDE (FR-1210, ICL), 83% Bromine
	Polymeric brominated FR, PolymBr, 62% solids dispersion, ~32% Bromine
	Polymeric brominated FR + phosphorus-containing species, PolymBrP, 50% solids dispersion, ~16% Bromine
Synergist	Antimony III oxide (ATO)
Cotton fabric	Woven, 250 g/m^2^

**Table 2 polymers-12-02160-t002:** Damaged lengths and depths of PU foams after flame extinction at the “pass” condition [45].

Formulation	Fabric Damaged Length, mm	Foam Damaged Depth, mm
DecaBDE/ATO	55	17
PolymBr/ATO	50	13
PolymBrP/ATO	195	20

**Table 3 polymers-12-02160-t003:** Properties of commercial grades of antimony III oxide and zinc stannates.

Antimony III Oxide, Sb_2_O_3_	Zinc Hydroxystannate (ZHS), Zn Sn(OH)_6_	Zinc Stannate (ZS), ZnSn0_3_
White powder	White powder	White powder
% Antimony 83.5	% Tin 41.0–43.0	% Tin 53.0–56.0
Water solubility 0.001 g/100 mL	% Zinc 22.0–23.5	% Zinc 26.2–27.5
water, 25 °C	% Moisture 0.7 max	% Moisture 0.5 max
Average particle size 0.4–1.8 microns	Average particle size (d50) 1.4–2.2 microns	Average particle size (d50) 1.4–2.2. microns

**Table 4 polymers-12-02160-t004:** Formulations and principal flammability parameters for tungstate/bromine-containing formulations in polyamide 6.6 (PA66) (adapted from reference [124]).

Sample	Composition (%)			Flammability Parameters			
	PA66	MC *	PolyBrFR	LOI, Vol.%	PHRR, kW/m^2^	TSR m^2^/m^2^	R_PHRR_ %
PA66	100	-	-	22.6	1644	609	-
BrPS	90	-	10	22.9	1049	1821	36.2
BrPBz	90	-	10	22.3	1206	1447	26.6
AlW **	95	5	-	23.0	1156	927	29.7
SnW **	95	5	-	21.5	954	939	42.0
ZnW **	95	5	-	22.0	1190	638	27.6
AlW-BrPS	85	5	10	23.3	999	1789	39.2
AlW-BrPBz	85	5	10	22.3	1174	1246	28.6
SnW-BrPS	85	5	10	26.7	546	1973	66.8
SnW-BrPBz	85	5	10	26.7	802	1766	51.2
ZnW-BrPS	85	5	10	26.2	485	949	70.5
ZnW-BrPBz	85	5	10	28.5	896	1186	45.5

Notes: BrPS = brominated polystyrene; BrPBz = poly(pentabromobenzyl acrylate); AlW, SnW and ZnW = aluminium, tin (II) and zinc tungstates respectively; * MC indicates each metal compound; ** values from [122]; PHRR is peak heat release; TSR is total smoke release; R_PHRR_ % is the percentage reduction in peak heat release rate, PHRR, with respect to PA66.

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
