# Peer review of "The Potential for Bio-Sustainable Organobromine-Containing Flame Retardant Formulations for Textile Applications—A Review"

_polymers, 2020, doi:10.3390/polym12092160_

Round 1

Reviewer 1 Report

The manuscript provides the Reader with a very well competently researched overview about the current state-of-the-art and some perspectives on the use of organobromine-containing flame retardant formulations for textiles with enhanced environmental sustainability. The references are properly selected and cited; the text is really well written and clear. Therefore I strongly recommend to publish it in the present form.

Author Response

No response required.

Reviewer 2 Report

The presents an thorough review of brominated flame retardant (FR) systems for textile back coatings, and I only have a few minor review comments. 

Page 1, lines 35-36:  Is it still correct that brominated FRs still have this much market share after mineral fillers like aluminum hydoxide?   Given how much halogen has been deselected, especially in the US where there is no longer any fire safety requirement for furniture, is this sentence still correct?   It may not be possible to validate this since market sales for FRs are closely held by the manufacturers. 

Line 40.  This should read "...toxicity of select organophosphorus-based retardants...."   Not all of them have been show to have negative toxicity profiles.  The reactive ones which covalently bond into epoxies and other systems are not under the same scrutiny as small molecule phosphorus compounds like triphenyl phosphate.  

Line 91.  Why are reactive brominated FRs not discussed here?  Brominated aromatic acrylates and tribromopentaerythritol are examples of reactive Br FRs that are being promoted to get away from the problem of small molecule Br FRs migrating out of materials. 

Line 123.  I Believe Saytex 8010 is produced by Albemarle Corporation, not MPI Chemie.  I would check this.  It may be produced under license or distributed by MPI Chemie, but Saytex 8010 is trademarked by Albemarle Corporation. 

Line 210-211.  I know what the author is trying to say about non-halogenated FRs not being as universally vapor phase as halogenated FRs, but this sentence really isn't correct.  There are phosphorus based FRs which can be vapor phase in the presence of certain polymers where the phosphorus species volatilizes rather than reacts with the decomposing polymer, melamine definitely has a vapor phase effect, and metal hydroxides can be argued to have some vapor phase effect (fuel dilution with water release) along with their obvious condensed phase effects (endothermic cooling, dilution of condensed phase fuel with non-flammable oxide).  I'd recommend rewording this sentence a bit to say that non-halogenated FRs require more tailoring / matching of FR chemistry with fiber/textile chemistry vs. the more general utilitarian nature of BrFRs. 

Lines 745-748.  I'd like to suggest the author include that focusing on these molecules as polymeric or reactive FRs is a path forward, rather than considering them as small molecule additives that can come off the back coating like pentabromodiphenyl ether did.  Brominated chemicals that are safe/natural to the marine environment may not necessarily be safe in a terrestrial environment.  While there are some definite biological similarities between land-based and water-based life, there are some notable chemical differences as well, and we should not assume that the molecules in Figure 4 will be safe to use especially since life in the sea is used to a halide-rich environment, and, has more halide available to incorporate into biomolecules than land-based life does.  I'll admit these chemical structures are very interesting and very likely to be effective FRs, but it would be better if they are locked into a polymeric structure, or covalently bound into polymers where they will maintain FR performance, rather than potentially migrating out into the environment.  Some tempering of this sentence is recommended. 

Author Response

See appended file.

Reviewer 3 Report

See attached

Author Response

See appended file.

Reviewer 4 Report

The manuscript is very detail and carefully prepared. I only have the following minor comments:

  • The so-called novel BFRs (such as BTBPE) in terms of their current usages and any potential environmental implications in reference to commonly deployed BFRs.
  • Are there any alternatives for Sb2O3? In a recent XRD analysis, we detected high loads of TbBr3 in incinerated printed circuit board, so we thought TbO2 might be deployed instead of Sb2O3.
  • It might be important to discuss the thermal stability limit of the investigated BFRs.

Author Response

Please find my responses below:

  1. The figures for relative usage of polymeric BrFRs are not available in the academic literature and their environmental impact has not been fully defined. See ref 8.
  2. The only commercial alternatives to ATO are the zinc stannates as discussed in Section 3.2.
  3. Currently used BrFRs are thermally stable to withstand typical processing temperatures. Their sensitivities to environmental thermal degradation have been reviewed in refs. 8, 82 and 83. Most mechanistic work has been undertaken at temperatures which simulate those during incineration and is rarely relevant to flame retardant chemistry where reaction temperatures are lower and involve synergist interactions as discussed in Section 3.

I hope I have addressed your concerns.